



# A Parquet Cube alternative to store gridded data for data analytics and modeling

Jean-Michel Zigna[1], Reda Semlal[1], Flavien Gouillon[3], Ethan Davis[4], Elisabeth Lambert[1], Frédéric Briol[2], Romain Prod-Homme[1,], Sean Arms[4], Lionel Zawadzki [3]

[1]Software Engineering Division, CLS, Toulouse, 31520, France
[2]Environmental Business Unit, CLS, Toulouse, 31520, France
[3]Earth Observation division, altimetry and radar department, CNES, Toulouse, 31400, France
[4]Division of the University Corporation for Atmospheric Research, UNIDATA, Boulder CO 80301, United States

*Correspondance to* : Jean-Michel Zigna (jzigna@groupcls.com)

**Abstract.** The volume of data in the field of Earth data observation has increased considerably, especially with the emergence of new generations of satellites and models providing much more precise measures and thus voluminous data and files. One of the most traditional and popular data formats used in scientific and education communities (reference) is the NetCDF format. However, it was designed before the development of cloud storage and parallel processing in big data architectures. Alternative solutions, under open source or under proprietary licences, appeared in the past few years (See Rasdaman, Opendatacube).

These data cubes are managing the storage and the services for an easy access to the data but they are also altering the input information applying conversions and/or reprojections to homogenize their internal data structure, introducing a bias in the scientific value of the data. The consequence is that it drives the users in a closed infrastructure, made of customized storage and access services.

The objective of this study is to propose a light new open source solution which is able to store gridded datasets into a native
big data format and make data available for parallel processing, analytics or artificial intelligence learning. There is a demand for developing a unique storage solution that would be opened to different users:

- Scientists, setting up their prototypes and models in their customized environment and qualifying their data to publish as Copernicus datatsets for instance;
- Operational teams, in charge of the daily processing of data which can be run in another environment, to ingest the
product in an archive and make it available to end-users for additional model and data science processing.

Data ingestion and storage are key factors to study to ensure good performances in further subsetting access services and parallel processing.

Through typical end users' use cases, four storage and services implementations are compared through benchmarks:

- Unidata's THREDDS Data Server (TDS) which is a traditional NetCDF data access service solution built on the
NetCDF-Java,
- an extension of the THREDDS Data Server using object store,
- pangeo/Dask/Python ecosystem,
- and the alternative Hadoop/Spark/Parquet solution, driven by CLS technical and business requirements.

## 1 Introduction

Thanks to the possibility to manage big and various datasets with velocity in big data architecture, there is an opportunity to quickly extract bigger data volumes, process them for new value-added products on larger geographical and temporal scales, with higher geographical and temporal resolutions, and provide more precise output data to build up new data science models intelligence from existing scientific data. For instance, in the animal monitoring field, scientists need to correlate locations, in-





situ observations with satellite or modelized environmental data on a long time period for the detection and the modeling of any new threat on the animal habitat due to Climate and Environment changes (reference EO4wildlife).

Cloud and bigdata technologies can help through their focus on horizontal scalability, by distributing the workload and I/O across a potentially resizable pool of resources. However, the effort to acquire new technical skills and to switch to this type of technology can be expensive and a barrier for scientists developing their data processing and artificial intelligence models

by their own.

Many oceanographic, climate, Earth observations datasets are in NetCDF format. This format is widely used to represent multi-dimensional arrays (see Unidata site). While the NetCDF format and the libraries ecosystem were initially designed for local, random-access files several approaches have been developed for enhancing the NetCDF libraries to make efficient use of cloud storage and parallel cloud processing. Support has also been added to the NetCDF-Java library to utilize HTTP byte-range

reading into NetCDF files stored in S3-compatible object stores to emulate random-access file reading. Using the new S3 capabilities of the NetCDF-Java library, the THREDDS Data Server (TDS) can provide data access services to S3-stored NetCDF files. Work has also be done for adding support to read and write Zarr data in the NetCDF core libraries (NetCDF-C and -Java). Preliminary results based on this initial THREDDS S3 work are presented in this paper.

Pangeo is already widely used by scientists in the geoscience field to process data in bigdata architectures using Python (see

https://speakerdeck.com/rabernat/pangeo-earthcube-tac-update?slide=13). The conversion of NetCDF data into Zarr data, based on NumPy, and the use of Dask are logical in the Pangeo ecosystem. In the industry and many firms, such as CLS, we can still find places where the development environment can be Python oriented, but the operational environment is based on Spark/Java/Scala technologies. Thus the idea is to find a common storage format and provide a unique datalake for both environments.

Several big data formats are candidate to extend the use of data in a Cloudera/Spark infrastructure. Cloud Optimized Geotiff, HSDS object store, Parquet: each of them has pros and cons, (reference https://ntrs.nasa.gov/citations/20200001178). Even if Parquet is not suitable for all datatypes, such as individual data points, it is relevant for gridded data. The question is to see how and to what an extent Parquet can be a viable and performant solution to promote. The different implementations tested in this publication to store and access to data are:

• THREDDS-NC for the traditional NetCDF environment,

• THREDDS-S3 for a new possibility using THREDDS services managing data stored in S3 storage (Amazon Simple Storage Service),

• Pangeo-Zarr for the Pangeo/Python/Zarr/Dask implementation,

• Spark-Parquet for the Cloudera/Hadoop/Spark/Parquet solution

**2 Methods**

To compare the existing THREDDS-NC, THREDDS-S3, Pangeo-Zarr and Spark-Parquet solutions, a selection of various datasets and use cases is realized.

For the datasets, the following criteria were identified to stress these implementations:

• Significant data volume: data is available over a one year duration,

• Variability in the geographical distribution: the geographical coverage is typically on the ocean or on the land,

• Significant set of variables to estimate the impact on the performances of the services.

For the use cases, we identified 3 typical scenarios:

• extraction of data in a geographical area over a time period for a subset of variables (3D or 4D) via a subsetter service,

• enrichment of CSV locations with variables values as additional CSV columns,

• Parallelized processing.





The selected datasets are open data from the Copernicus Core Services. The modelized collection GLOBAL_ANALYSIS_FORECAST_PHY_001_024, from CMEMS, is covering the oceans, big continuous surfaces, with a diversity of variables and time steps resolution (see a short description of tested data in table 1 and for a complete description

https://resources.marine.copernicus.eu/?option=com_csw&view=details&product_id=GLOBAL_ANALYSIS_FORECAST_ PHY_001_024).

The lake water quality product, https://land.copernicus.eu/global/products/lwq, from CLMS, is made of small geographical spots over European and African regions.

The Sea Level Anomalies dataset, http://marine.copernicus.eu/services-portfolio/access-to-

products/?option=com_csw&view=details&product_id=SEALEVEL_GLO_PHY_L4_REP_OBSERVATIONS_008_047 , drives the reuse of an existing Pangeo notebook to illustrate the parallel processing: https://github.com/pangeo-gallery/physical-oceanography/blob/master/01_sea-surface-height.ipynb.

| dataset | Number of variables | Files | Time coverage | Size (Tb) |
|---|---|---|---|---|
| CMEMS Global Analysis Forecast daily | 11 variables | 1 file per day, 1 time step 366 files | 1 year rolling online data | 1.22 |
| CMEMS Global Analysis Forecast hourly | 4 variables: Mix 3D and 4D) : Ssh, u,v, temp | 1 file per day, 24 time steps 366 files | 1 year rolling online data | 0.578 |
| LWQ-100 | N variables | 1 file per 10d 36 files | 1 year of data | 0.068 |
| SLA | N variables : sea_surface_height_abo ve_sea_level (SSH) for sea level rise use case | 1 file per day 9625 files | Since 1993 | 0.070 |

**Table 1: tested datasets summary**

All these datasets are available as NetCDF files. According to each implementation, datasets can be converted into another format to optimize the execution time of the scenarios.

The first scenario is made of a set of 100 requests to a subsetter service. It is defined in order to run and replay the same extractions for each implementation.

Storage size for inputs and outputs of the requests and execution times of the subsetter service are discussed in the results item

for each implementation.

For the second scenario, an enrichment along tracks is performed. A set of additional variables, growing in numbers, is added to all positions in the tracks, over a specific region or worldwide. To get a significant number of real tracks, anonymized ships locations were exported from the Automatic Identification System archive available at CLS premises. During the enrichment, a linear interpolation using the surrounding values was performed. The implemented interpolation is simply a weighted

arithmetic mean of the neighbours' values, where the weight of each value corresponds to the inverse of the distance (in meters) between the neighbour and the track point.

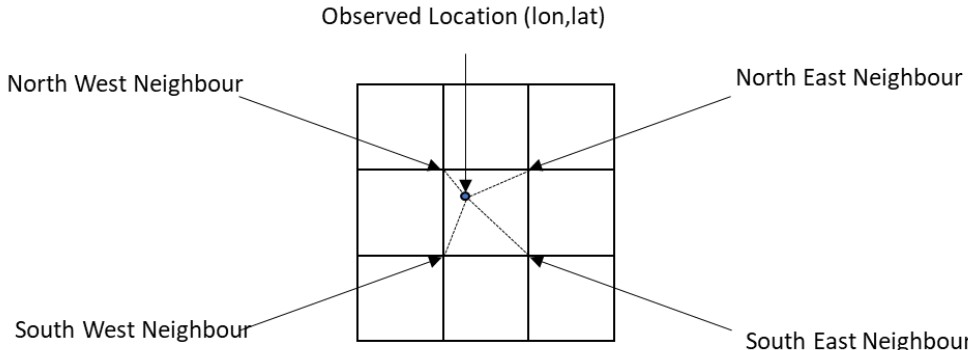

**Figure 1: interpolation along track**

For the third scenario which is parallelized processing, the CMEMS sea level anomalies product is used to compute and display the yearly mean sea surface elevation since the beginning of altimetry measurement in 1993.

Each implementation is detailed below to illustrate its storage mechanism, its services for geographical extractions or along track enrichment, and its limits and potential workarounds. Then the results are illustrated to provide an overview of their
performances.

## 2.1 Subsetter scenario 1 details

**Run**: For each request, generate an extraction including all required variables according to the geographical coverage over the expected time period of interest, as a single NetCDF file or in another format in line with the solution. To write NetCDF 4
output with librairies, the chunk size is set to 256 kB, Other parameters are default values.

**Metrics**
- Global execution time: on a hot instance (service initialized, up and ready to request)

For the marine dataset, the criteria of subsetting are:
- A list of variables
- Areas, defined as rectangles (top left, right down corners in lon, lat):
    - North Sea: small size to check the incidence of filled value due to shore areas: [lon(-10,10),lat(50,60)]=$100°^2$
    - Atlantic Ocean: small size (4 times the surface defined for the North Sea): [lon(-30,10),lat(50,60)]= $400°^2$
    - Pacific Ocean: medium size with check of 180° management (that can required 2 extracts to merge):
[lon(140,200), lat(0,35)]=$2100°^2$
    - Global coverage: [lon(-180,180),lat(-90,90)]=$32400°^2$
- Time depth: from 1 month to 12 months

For the land dataset, the criteria of subsetting are:
- 1 variable: Turbidity_blended_mean
- Time coverage:
    - 1 day: 2020-02-29
    - 1 month: 2019-09-01 to 2019-09-30,
    - 3 months: 2019-09-01 to 2019-11-30,





- ○ 6 months: 2019-09-01 to 2020-02-29,
- ○ max: 2019-04-01 to 2020-03-31
- • Spatial coverage:
- ○ zone 10 ° X10 ° on equatorial Africa [(30,0),(40,-10)],
- ○ Europe + Africa: [(-20,70),(50,-70)]

A first set of requests is performed to try to assess the incidence of the dataset characteristics on subsetting results and estimate the limits of the implementation due to the time coverage or number of variables to extract. The system configuration for these extractions is a Linux server with 10 cores and 4G of memory.

A second set of requests is performed to see how the number of cores, for a given dataset and a given request, is impacting the subsetting performances and how scalable the solution is. The number of cores is increased for the Pangeo-Zarr and the Spark-
Parquet environments.

All the requests are described in an excel file available with the tested datasets (see Data Availability).

**2.2 Enrichment along track scenario 2 details**

**Run**: extraction of environmental data along tracks (set of positions, Input in csv, output in csv)

**Metrics**:
- • Global execution time: on a hot instance

The parameters that are used to stress the solutions are:
- • Number of points of the track
- • Zone: North Sea, Rotterdam, globe to estimate the geo coverage factor
- • Time depth: from 1 day to 1 year
- • Number of cores
- • Number of variables to extract

A first set of requests is performed to evaluate the incidence of the geographical coverage and of the number of positions.

A first AIS dataset includes tracks on the area of North Sea and Rotterdam: (the filename contains the down sampling time
period)
- • NorthSea.csv : 3 207 817
- • Rotterdam1H.csv : de 05/04/2019;00:00:00 à 05/04/2019;23:59:59 : 152 484 positions
- • Rotterdam15m.csv : de 06/04/2019;00:00:01 à 06/04/2019;23:59:59 : 362 557 positions
- • Rotterdam12H.csv : de 02/04/2019;00:00:00 à 02/04/2019;23:59:58 : 76 764 positions
- • Rotterdam1m.csv : de 03/04/2019;00:00:00 à 03/04/2019;23:59:59 : 1 025 641 positions

RotterdamAll.csv : de 07/04/2019;00:00:00 à 07/04/2019;23:59:59 : 1 638 373 positions

A second AIS dataset is set up to estimate the time period incidence on a worldwide distribution:
- • World1 (all positions of all vessels for 1 day) : 2 578 627 positions
- • World1D (one position by vessel for 1 day) : 176 735 positions
- • World5D (one position by vessel by day for 5 days): 876 941 positions
- • World10D: 1 736 849 positions
- • World20D: 3 588 894 positions
- • World50D: 8 897 349 positions






Then, the incidence on the number of extracted variables and of the number of cores to enrich the location are also studied.

### 2.3 Parallel processing scenario 3 details

**Run**: Calculate the annual average elevation of the oceans on the globe over 25 years

- Fill value to ignore
- Output : curve plot (png)
- Conf : 4 go per executor, 5 cores per executor (optimum), up to 50 cores

**Metrics**:

- With the initialization time of the cluster to have an significant estimation from the end user point of view

With sea level anomaly variable, draw on the same plot:

- The average by time step of the values
- The sliding average with a window size of 365 days

### 3 Description of implementations

### 3.1 THREDDS implementation

Unidata's THREDDS Data Server (TDS) provides a variety of data access services that can be used to serve NetCDF datasets.
The NetCDF Subset Service (NCSS) supports coordinate system requests and can return the requested data in several formats including NetCDF-4. Unidata's TDS uses the NetCDF-Java library to read the datasets to be served by the TDS. This is the THREDDS implementation.

By extending the NetCDF-Java library to support byte-range access to object stores that support the S3 API, the existing TDS services can now be used to serve NetCDF files stored as objects in an object store. It is the THREDDS-S3 implementation.
While this new S3 access can be used to serve individual NetCDF files, extending the new S3 capability to TDS aggregation of datasets is still being implemented. This means that the year of daily files for the two CMEMS hourly and daily mean dataset collections cannot be served as a virtual aggregation of the existing files but must instead be concatenated into a single file per dataset collection. This concatenation was done using the NCO `ncrcat` command.

TDS services (including NCSS) were designed for targeted subset data access and were not designed to handle requests that result in very large responses. The size of results from some of the benchmark requests resulted in responses that exceeded size limits in place on the systems. The perhaps related second issue involved requests reaching timeout limits.

While the TDS handles multiple requests in a multi-threaded manor, individual requests are not currently handled in a multi-
threaded manor. Working to parallelize reads to S3 objects is something on the THREDDS list for future work (see list below).

To work around the size and timeout limitations, some of the original requests were modified by adding horizontal and/or temporal strides to the requests thus reducing the volume of the resulting data. Adding a stride means that not every data point in the requested spatial/temporal extent is returned. Instead only every n-th data point along the spatial dimensions or temporal
dimension are returned, where 'n' is the value given by the stride parameter.

### 3.2 Pangeo implementation

Pangeo is, first and foremost, a community of people working collaboratively to develop software and infrastructure for geoscience research on large volume of data. Among the products developed by this community, there are interconnected





software that can be deployed in cloud computing or high-performance computing (HPC) environments. Such a deployment is also known as a "Pangeo environment."

These environments are built around a software stack using the Python computer language and reference libraries, allowing large-scale processing. The central element of this ecosystem is the Dask library. This library enables to run calculations on a distributed or local cluster. Dask uses task graphs and data structures to distribute tasks and memory. Data structures are

handled in memory by Dask. Third-party libraries such as Zarr takes care of reading and writing data to the disk. Data processing libraries, such as Dask, manage data internally by dividing it into smaller chunks to enable distributed access, efficient compression and to reduce input/output access time on the file system.

In this implementation, the conversion of the different NetCDF datasets into Zarr format is performed by a specific Python script based on the NetCDF4 module. The written data is compressed with the LZ4 algorithm using a compression level of 5.

When relevant for certain physical variables, filters (ex., delta, scale factor, add offset) are added to improve the compression rate of these variables.

### 3.3 Parquet Cube implementation

The development of the Parquet Cube solution is driven by the following scientists' requirements:

1. Do not alter the NetCDF input quality and precision
2. Make it simple, generic, robust and scalable
3. Allow further large scale processing for ML or DL models
4. Be compliant with traditional scientists' habits (having everything on their computer)
5. Open the use of the solution to a large community of users

For item 1: Existing data cubes are ingesting data to be able to process them in the same projection, if needed the same geospatial resolution 'see open cube ingestion'. Scientists want to keep the original information or have a complete description of the ingestion mechanism, which is not clear once data is available through the data cube access services. So reprojections, or conversion of units, interpolations are prohibited in this solution. This introduces a limitation to manage X,Y geographical representation used in polar datasets for instance. The single recommendation in place is to homogenise the precision in latitude

and longitude between datasets to be able to use such latitudes or longitudes key values during select or join operations. The decimal precision is set to 0.00001 degrees, representing approximately a 1-meter precision at the Equator line. Depending on the variable type defining longitude and latitude, this rounding operation is also useful to avoid floating point error and failing jointures during requests.

For item 2: a scalable docker ingestion process is in place to deal with inputs of GBytes and thousands of files.

For item 3: Parquet is a common big data format standard. Several libraries, In Java, Scala, Python or R, familiar to data scientist and educational communities, are reading it. Final storage can be done through additional local storage disks, HDFS or object storage like S3 buckets for big volumes available in big data infrastructures. Be aware that HDFS replicated storage

can be typically three times the original size of a file. The operational storage costs are not studied here.

For item 4: The proposed parquet cube solution can be the input format of a scalable subsetter service, to extract data for scientists matching their geographical, time coverage and environmental variables list of interest. The output of the subsetter service can load data in memory for further parallel processing or can generate a downloadable output file for any scientist

who wants to process data directly on his computer. The first option is to write the output as a NetCDF file, but the storage





and the time to process the output are quickly prohibitive. The second option is to benefit from the scalability of the bigdata enabled data format such as Parquet as an output format. The results of the two options are described in the "Results" item.

For item 5: the conversion from NetCDF to Parquet is done by using an opensource software. The objective is to open the source code to users for any further optimization or modifications required to ingest their NetCDF inputs.


To meet the above requirements, the Parquet Cube implementations is using the following ingestion mechanism. The overall idea is to store each data point from the original gridded data as a row in a Parquet table. Thus, each row has columns corresponding to NetCDF dimensions and to variables values.

|  | Dimensions | | | | Variables | | | |
|---|---|---|---|---|---|---|---|---|
| Latitude | Longitude | Time | Depth | Var 1 | Var i | Var j | Var N |
|  |  |  |  |  |  |  |  |
|  |  |  |  |  |  |  |  |


**Figure 2: Flattening a multi-dimensional array in tabular format**

In addition to this parquet file, two separate files are generated:

- A dimension index file: An index for each dimension (one file per dimension).
- 275 A metadata file (a single file per dataset) in json format, an instance is created for each new version of the Parquet structure.

This metadata file stores all the attributes extracted from the original NetCDF files of the dataset. This can be useful to generate a NetCDF output in line with the NetCDF input but also to implement a description service of the dataset (not described here).

```
prod/data/metoc/datasets/Altimetry_Sea_Water_Velocity/dataset-alti8-nrt-global-msla-uv
|----------data
|------------tspartday=2019001
|---------------dataset-alti8-nrt-global-msla-uv.20190101.snappy.parquet
|------------tspartday=2019002
|---------------dataset-alti8-nrt-global-msla-uv.20190102.snappy.parquet
|------------tspartday=2019003
|---------------dataset-alti8-nrt-global-msla-uv.20190103.snappy.parquet
|------------tspartday=2019004
|---------------dataset-alti8-nrt-global-msla-uv.20190104.snappy.parquet
|------------tspartday=2019005
|---------------dataset-alti8-nrt-global-msla-uv.20190105.snappy.parquet
|------------tspartday=2019006
|---------------dataset-alti8-nrt-global-msla-uv.20190106.snappy.parquet
|------------tspartday=2019007
|---------------dataset-alti8-nrt-global-msla-uv.20190107.snappy.parquet
|------------tspartday=2019008
|---------------dataset-alti8-nrt-global-msla-uv.20190108.snappy.parquet
|------------tspartday=2019009
|---------------dataset-alti8-nrt-global-msla-uv.20190109.snappy.parquet
|------------tspartday=2019010
|---------------dataset-alti8-nrt-global-msla-uv.20190110.snappy.parquet
|------------tspartday=2019011
|---------------dataset-alti8-nrt-global-msla-uv.20190111.snappy.parquet

---------------dataset-alti8-nrt-global-msla-uv.20201020.snappy.parquet
|------------tspartday=2020295
|---------------dataset-alti8-nrt-global-msla-uv.20201021.snappy.parquet
|------------tspartday=2020296
|---------------dataset-alti8-nrt-global-msla-uv.20201022.snappy.parquet
|----------index
|------------latitude.lst
|------------longitude.lst
|------------metadata.json
|------------metadata.json.1
|------------metadata.json.2
|------------metadata.json.3
|------------metadata.json.4
|------------metadata.json.5
|------------metadata.json.6
|------------time.lst
```


**Figure 3: dataset storage in Parquet**





Due to the size of the tested input files and some typical end user requests, the partitioning-per-day makes sense The partitioning key is calculated for each data point from the value of the time dimension in the format YYYYDDD (day of year). This allows to benefit from quick partition pruning based on dates, since almost all our use cases are applied to a pre-defined time interval.


During the data transformation from NetCDF to parquet format, the following basic transformations are performed to avoid additional operations in the subsetter service:

- Normalization and rescaling of latitude/longitude values (latitudes are rescaled to the range [-90, 90] and longitudes to the interval [-180, 180]). This enables the ability to join several datasets on the latitude/longitude without
considering the ranges.
- Time dimension values are converted to unix epoch timestamps, to avoid disparities between datasets.
- Variable values read from NetCDF files are rescaled using the offset and the scale factor declared in the NetCDF file metadata, to allow comparisons of values between datasets (this would otherwise needs calculating the real values during the processing).
- It is to be noted that we implement a special handling for dataset that mix 3d and 4d variables: Since all variables, regardless of their dimensions are grouped in the same parquet table, data is stored as if the depth dimension was nullable and all variables were 4d variables.
- the following example illustrates how the parquet table looks like in this specific case: The 4d variables are considered as "null" (empty cell) when the depth is "null", and the 3d variables are considered as "null" when depth is not.

For the tested datasets, the Parquet storage is designed to get a daily parquet file per folder to obtain a timely indexation, with a storage size compliant with good I/O performances. The compression format is Snappy since it is splittable (see Cloudera documentation), performant in read and write access for any dataset.

Additional metadata columns are added in the footer of the Parquet format to quickly filter data between physical values ranges for instance (see Faster Performance for Selective Queries).


| Latitude | Longitude | Time | Depth | Var 1 | Var i | Var j | Var N |
|---|---|---|---|---|---|---|---|
| Lat0 | Lon0 | T0 | | V13d(T0,Lat0,Lon0) | Vi3d(T0,Lat0,Lon0) | Vj3d(T0,Lat0,Lon0) | Vn3d(T0,Lat0,Lon0) |
| Lat1 | Lon1 | T1 | | V13d(T1,Lat1,Lon1) | Vi3d(T1,Lat1,Lon1) | Vj3d(T1,Lat1,Lon1) | Vn3d(T1,Lat1,Lon1) |
| ... | | | | | | | |
| Lat0 | Lon0 | T0 | d0 | V14d(T0,Lat0,Lon0,d0) | Vi4d(T0,Lat0,Lon0,d0) | Vj4d(T0,Lat0,Lon0,d0) | Vn4d(T0,Lat0,Lon0,d0) |
| Lat1 | Lon1 | T1 | d1 | V14d(T1,Lat1,Lon1,d1) | Vi4d(T1,Lat1,Lon1,d1) | Vj4d(T1,Lat1,Lon1,d1) | Vn4d(T1,Lat1,Lon1,d1) |
| ... | | | | | | | |

**Figure 4: Dataset mixing 3D and 4D variables**

The Parquet format implements several compression methods. Here below are some metrics in I/O operations performed during this study, in line with the literature:

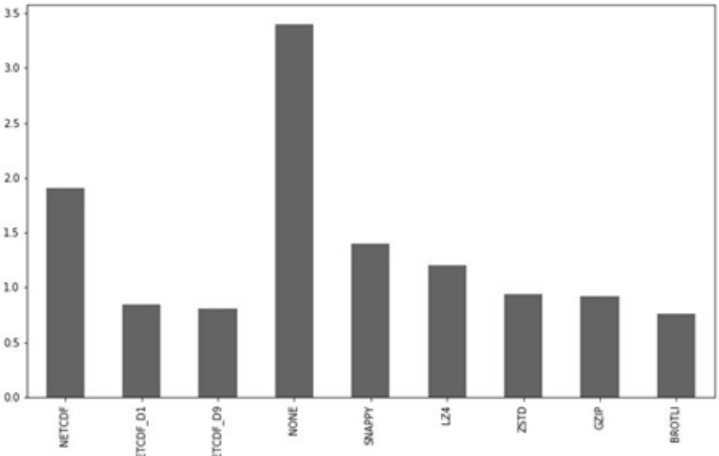


**Figure 5: Storage size by compression method (in Gb)**

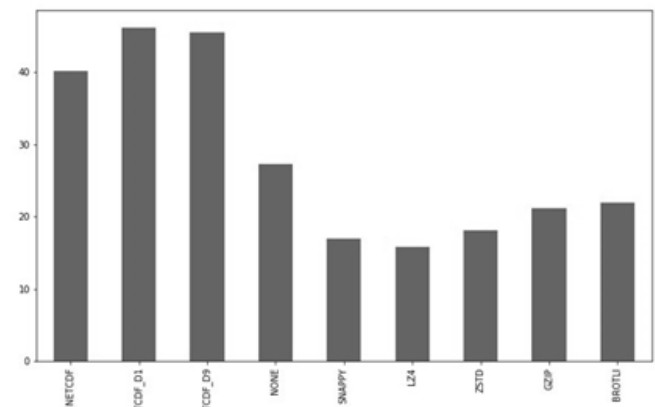

**Figure 6: read access time (in seconds)**

The parquet files structure is stored on HDFS, and Apache Spark is used to read and process data from parquet files in a distributed manner. This Parquet storage can be accessed through a dedicated subsetter service or several technologies like Spark or Python Dask notebooks.

As a reminder, value subsetting consists in queries that select only data points for which the associated value is bounded by a

predefined interval. Parquet metadata are used to select chunks of interest while running value range search and performing scans only on the chunks for which the interval [min, max] intersects with value range of interest.

Note that the subsetter service used in the trials is under CLS property intellectual rights, currently not exposed or distributed.



**4 Results**

**4.3 Execution environments**

The Hardware architecture for each solution is described below, all running on UNIX Virtual machines.

THREDDS implementation is using two TDS instances. One accesses the single file benchmark datasets from local disk. The second accesses the single-file benchmark datasets from an on-prem object store appliance. The two TDS instances provide us with an interesting comparison for the benchmark testing.

The TDS service manages data stored on a local file system is a dual Xeon Gold 6138 machine with 512GB memory and a 120TB RAID running CentOS. The TDS that serves data stored on an object store, is a VM on a cluster co-located with the object store, it has 4 CPUs with 12 GB of RAM and 80 GB disk and is also running CentOS.

The CNES HPC cluster was used to deploy Pangeo on a Conda environment. PBS Pro drives this cluster, and GPFS operates the distributed file system. The main libraries used with our Pangeo environment are Dask, Dask-jobqueue, Zarr, and Pyinterp.

Pyinterp was used to perform the interpolations. For each test, PBS was used to limit the resources allocated on the cluster. We repeated the tests several times to obtain the most reliable metrics. Nodes in the CNES HPC have the following characteristics:

- CPU: 24 cores / 2.2 GHz (Intel Broadwell)
- Memory: 128 Go / 2400 MHz
- Disk storage : 2 x SAS 15Klaps/mn / 300 Go / RAID0 / 500 Go
- Network : Gigabit Ethernet
- Infiniband : FDR 56Gb

The Parquet Cube solution is deployed on the CLS big data infrastructure (Cloudera version 5.13.1).




**Figure 7: Parquet Cube ingestion and processing diagram**

The ingestion processing produces the parquet files consumed later by Spark jobs to extract data through Jupiter or Zeppelin Notebooks or through APIs such as the subsetter service.


Unfortunately, within the scope of this study, it was impossible to deploy each solution on the same infrastructure. To try to minimize the impact of the hardware architecture, performances for executing the requests were measured with the same number of CPU and memory size, but with different disk storage performances.





### 4.3 Comparison of results of each solution

**4.3.1 Ingestion of NetCDF data**

The first THREDDS deployment is using the current NetCDF format, so no conversion is required. The second one is using the S3 storage of a unique object per dataset. The conversion uses the NCO `ncrcat` command to generate a single object store. Concerning Pangeo-Zarr solution, a python script was developed using the xarray module to convert NetCDF into Zarr format. XARRAY module was not used to convert the big volume of the datasets and a specific script was developed using NetCDF4

Python module for input read and using Dask/Zarr for output write.

The ingestion service for the Spark-Parquet conversion is implemented as a Java/Scala job encapsulated in a docker (see git repository for a complete description).

| Dataset | NetCDF | THREDDS-S3 | Zarr | Parquet |
|---|---|---|---|---|
| CMEMS Global Analysis Forecast daily | 1.22 | 1.388 | 0.454 | 0.720 |
| CMEMS Global Analysis Forecast hourly | 0.578 | 0.620 | 0.230 | 0.129 |
| CLMS Lake Water Quality | 0.068 | N/A | 0.006 | 0.164 |
| CMEMS Sea Level Anomalies | 0.070 | N/A | N/A | 0.074 |

**Table 2: Size of datasets in Tb**

For big datasets, we can see the impact of the compression method and of the replication of data. There is no replication for the Pangeo-Zarr or THREDDS implementations. It is to consider that HDFS storage is replicating 3 times the data chunks to monitor the disaster recovery data integrity (so the effective disk storage is 3 times the presented figures). For the Lake Water Quality, the ingestion size in the Parquet format could be optimize by the generation of a second dataset with variables first_observation_day, last_observation_day, number_of_observations which are metadata and not observations. This can be

discussed according to the usage of the lake water quality physical measurements.

| Dataset | THREDDS-S3 | Zarr | Parquet |
|---|---|---|---|
| CMEMS Global Analysis Forecast daily | 1328 | N/A | 1223 |
| CMEMS Global Analysis Forecast hourly | 581 | N/A | 426 |
| CLMS Lake Water Quality | N/A | N/A | 12816 |
| CMEMS Sea level anomalies | N/A | 1 | 3.6 |

**Table 3: Single file Conversion CPU time in seconds**

The Zarr conversion on the HPC was not measured precisely, but using all the expected nodes in the CNES HPC, the ingestion of all the datasets took approximately 1 day. In comparison, the Parquet ingestion for all the datasets was 7.4 days on the CLS

infrastructure without parallelization.

### 4.3.2 Subsetting access results

The first idea concerning the implementation of the scenario was to generate NetCDF outputs to provide extracted data as a familiar format to scientists. But limits were found, both in output file size and execution time to generate the NetCDF result. The NetCDF output was not generated in the end when the output size was reaching the limits of the tested implementation.

A workaround was put in place for each implementation.





For THREDDS, since the output is NetCDF, a stride strategy is put in place to limit the output of the subsetting to a threshold of 600Gb compliant with NCSS. Some requests, even with stride values given, still resulted in timeout issues or out of memory issues in the case of S3 access. Changes of the settings on how TDS caches S3 reads solved some out of memory issues. Remaining out of memory issues are believed to be related to how data is handled when generating the resulting NetCDF-4 files. These issues have caused the TDS testing not having results for all the benchmark requests which are identified as N/A below.

For the Pangeo-Zarr and the Spark-Parquet implementations, it was initially decided to generate NetCDF outputs when their size was less than 1 Tb, and then to generate Zarr and Parquet outputs respectively to benefit from the scalability of the architecture.

To refine the estimation of the execution times, requests were run 5 times for Pangeo-Zarr, 3 times for Spark-Parquet. The mean result is presented below.

**Incidence of geographical and time coverage**

The sea_surface_height_above_geoid variable of the global-analysis-forecast-phy-001-024 dataset was extracted (for Pangeo-Zarr and Spark-Parquet using 10 cores of 4 Gb).

| Area | Time | THREDDS-NC | THREDDS-S3 | Pangeo-Zarr | Spark-Parquet |
|---|---|---|---|---|---|
| North Sea | 1D | 1,80 | 4,66 | 0,40 | 5,00 |
| North Sea | 1M | 3,17 | 16,63 | 0,27 | 6,00 |
| North Sea | 3M | 7,52 | 39,88 | 0,35 | 8,00 |
| North Sea | 6M | 12,99 | 74,33 | 0,55 | 12,00 |
| North Sea | 1Y | 24,84 | 141,86 | 1,37 | 17,00 |
| Atlantic Ocean | 1D | 0,34 | 4,27 | 0,08 | 5,00 |
| Atlantic Ocean | 1M | 0,73 | 16,11 | 0,25 | 6,00 |
| Atlantic Ocean | 3M | 2,95 | 39,19 | 0,30 | 9,00 |
| Atlantic Ocean | 6M | 4,72 | 72,73 | 0,49 | 11,00 |
| Atlantic Ocean | 1Y | 7,86 | 139,70 | 1,28 | 16,00 |
| Pacific Ocean | 1D | 0,82 | 6,75 | 0,10 | 4,00 |
| Pacific Ocean | 1M | 10,41 | 71,92 | 0,70 | 7,00 |
| Pacific Ocean | 3M | 22,32 | 190,93 | 0,64 | 10,00 |
| Pacific Ocean | 6M | 39,74 | 387,95 | 1,00 | 11,00 |
| Pacific Ocean | 1Y | 86,83 | N/A | 1,86 | 20,00 |
| Global | 1D | 3,54 | 12,14 | 0,11 | 6,00 |
| Global | 1M | 86,44 | N/A | 0,91 | 12,00 |
| Global | 3M | 328,95 | N/A | 1,79 | 23,00 |
| Global | 6M | 280,48 | N/A | 2,95 | 36,00 |
| Global | 1Y | 408,56 | N/A | 5,70 | 68,00 |

**Figure 8: 3D Subsetting in time (seconds) and area of interest**

We can see that the execution time is quite linear between 1 month and 6 months for THREDDS-NC and THREDDS-S3 with a coefficient equal to 1 for a given area, but due to the striding strategy, it is not possible to conclude on a linear factor for the geographical dimension.





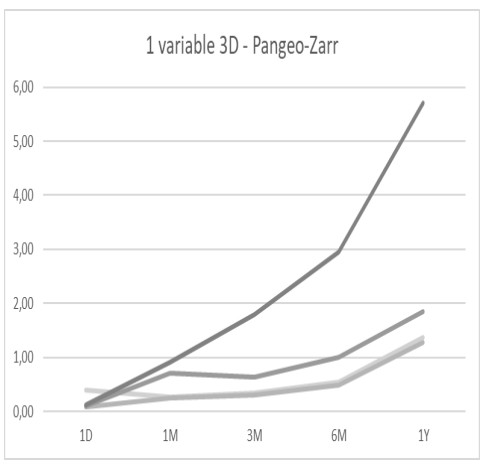 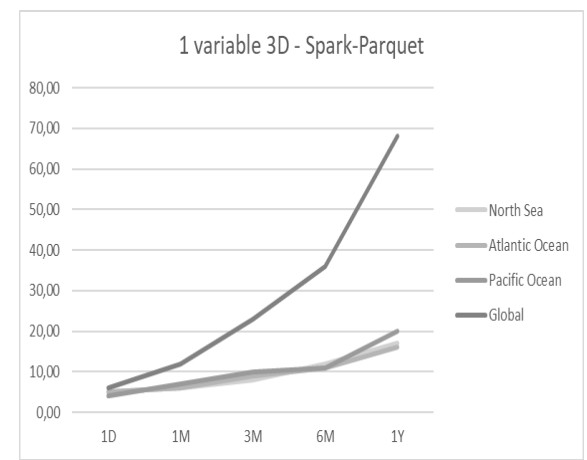

**Figure 9: 3D Subsetting time in seconds**

We can see a quadratic behaviour for the extractions performed with Pangeo-Zarr and Spark-Parquet. Pangeo-Zarr has a higher sensitivity to area size than Spark-Parquet for which extraction times are in the same order except for the global coverage.

Even If the execution times are around 1 minute for the global extraction over one year for Spark-Parquet in the CLS infrastructure, they are around 10 times slower than the Pangeo-Zarr results in the HPC data centre for the Global extraction, up to 60 times slower for some local areas.

**Incidence of the number of dimensions**

The 4d sea_water_potential_temperature is extracted for depth 5.078224.

| Area | Time | THREDDS-NC | THREDDS S3 | Pangeo-Zarr | Spark-Parquet |
|---|---|---|---|---|---|
| North Sea | 1D | 0,58 | 4,46 | 0,23 | 4,00 |
| North Sea | 1M | 2,51 | 15,14 | 0,55 | 6,00 |
| North Sea | 3M | 7,52 | 36,35 | 1,28 | 15,00 |
| North Sea | 6M | 14,18 | 67,55 | 2,35 | 19,00 |
| North Sea | 1Y | 19,73 | 127,34 | 4,46 | 32,00 |
| Atlantic Ocean | 1D | 0,44 | 4,08 | 0,19 | 8,00 |
| Atlantic Ocean | 1M | 0,80 | 14,45 | 0,51 | 13,00 |
| Atlantic Ocean | 3M | 4,39 | 35,08 | 1,26 | 12,00 |
| Atlantic Ocean | 6M | 10,45 | 65,38 | 2,32 | 18,00 |
| Atlantic Ocean | 1Y | 10,96 | 126,56 | 4,47 | 35,00 |
| Pacific Ocean | 1D | 0,53 | 5,72 | 0,28 | 5,00 |
| Pacific Ocean | 1M | 10,36 | 34,68 | 1,04 | 11,00 |
| Pacific Ocean | 3M | 23,04 | 90,66 | 2,80 | 21,00 |
| Pacific Ocean | 6M | 28,81 | 191,02 | 5,35 | 17,00 |
| Pacific Ocean | 1Y | 124,73 | N/A | 10,51 | 36,00 |
| Global | 1D | 5,04 | 13,60 | 0,23 | 5,00 |
| Global | 1M | 85,34 | N/A | 2,20 | 22,00 |
| Global | 3M | 278,61 | N/A | 5,78 | 31,00 |
| Global | 6M | 237,13 | N/A | 11,11 | 48,00 |
| Global | 1Y | 376,83 | N/A | 22,26 | 88,00 |

**Figure 10: 4D subsetting in time (seconds) and area of interest**

Compared to 3D extractions, we can see a significant increase in time for 4D requets for the Pangeo-Zarr and the Spark-Parquet implementations: 3 times slower for Pangeo-Zarr, only a half more for Spark-Parquet.

This is not the case for the THREDDS implementation because of the stride policy put in place to be able to extract data in the THREDDS NCSS file size limit which is 600MB. Striding means that a data point is extracted every N along a dimension (N can be between 2 to 20 depending on the request).



**Incidence of the number of variables**

The requests to study the impact of the number of variables are based on the following variables:

420         1: sea_surface_height_above_geoid

         4: sea_surface_height_above_geoid,eastward_sea_water_velocity,

northward_sea_water_velocity,sea_water_potential_temperature

         11: sea_water_salinity,sea_water_potential_temperature_at_sea_floor,ocean_mixed_layer_thickness_defined_by_si

gma_theta,sea_surface_height_above_geoid,eastward_sea_water_velocity,northward_sea_water_velocity,sea_water

425         _potential_temperature,sea_ice_area_fraction,sea_ice_thickness,northward_sea_ice_velocity,eastward_sea_ice_vel

ocity

As before, the requests are processed by combining:

- a given area: North Sea, Atlantic Ocean, Pacific Ocean, Global
- A period of time: from the lightest to the darkest line : 1 day, 1 month, 3 months, 6 months, 1 year

430        **Figure 11: incidence of the number of variables on the execution time in seconds**




The THREDDS execution times are slower than the ones in the Pangeo-Zarr or the Spark-Parquet implementations. The THREDDS implementation is also including the stride bias. For Pangeo-Zarr and Spark-Parquet, we can see that the execution time is less sensible to the number of variables for small areas and becomes more linear for the global coverage. The Pangeo-Zarr implementation on the CNES HPC can retrieve data faster than the CLS Spark-Parquet implementation but depends more

on the number of variables.

The THREDDS-S3 implementation did not provide all results to be able to make an interpretation of the results.

**Incidence of number of cores**

One of the key factors in the execution time is the number of cores to process data. The diagram below illustrates the results
for the extraction of sea_water_potential_temperature, eastward_sea_water_velocity,northward_sea_water_velocity, sea_surface_height_above_geoid variables for the global-analysis-forecast-phy-001-024 dataset over a 3 months time period.

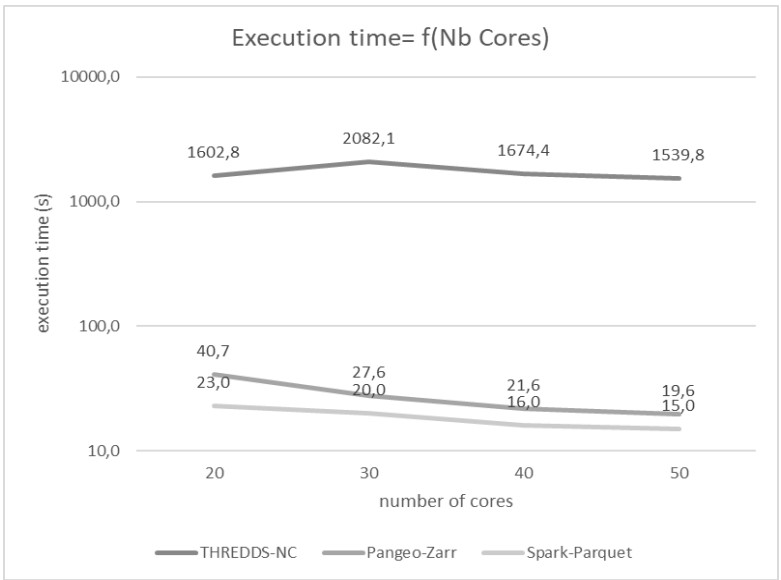

**Figure 12: incidence of number of cores**


The optimum number of cores is 40: the limitation of the execution time for greater values of number of cores is not linked to the CPU and memory capacity but limited by the I/O performances.

**Incidence of the number of time steps**

The test is to assess if the number of time steps in a dataset is a key factor in terms of performances and to try to find out what the limits are in the different implementations. The request is the same: 4 variables, the sea_water_potential_temperature, eastward_sea_water_velocity, northward_sea_water_velocity, sea_surface_height_above_geoid, extracted first from the global-analysis-forecast-phy-001-024 daily dataset and then from the global-analysis-forecast-phy-001-024-hourly-t-u-v-ssh the hourly dataset having 24 more time steps than the daily. The size of the first one is approximately twice as the second one
since it includes more variables. The THREDDS-S3 implementation provided a single value for the hourly dataset and is not considered in the analysis.



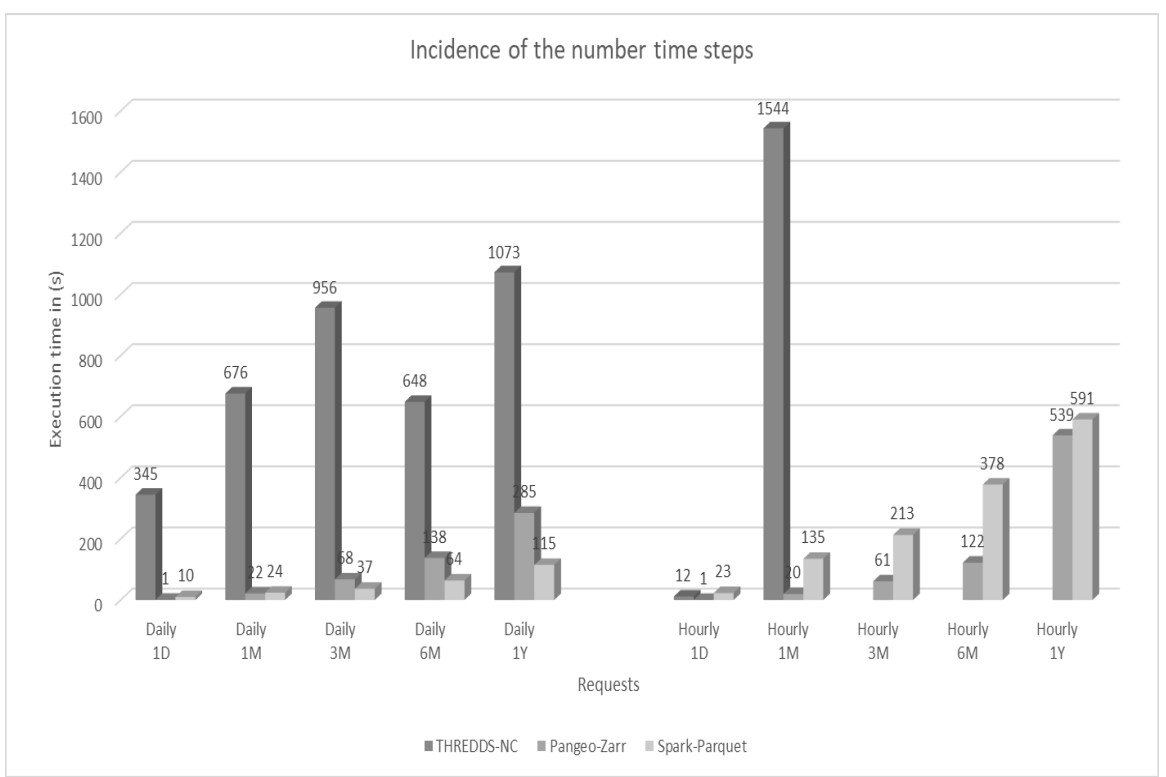

**Figure 13: incidence of the number time steps**

The number of time steps is impacting greatly the results, but Pangeo and Parquet implementation can extract data with a factor which is not correlated to the number of time steps but to the extracted duration. For a given time period, the execution time for the extraction of the hourly dataset is not 24 times the execution time of the daily dataset. it is because the access is driven by an indexation in days and not in hours.






### Incidence of not significant data

The idea was to compare the incidence of the data type: significant continuous data over wide areas (seas) versus lot of small data spots (lakes).

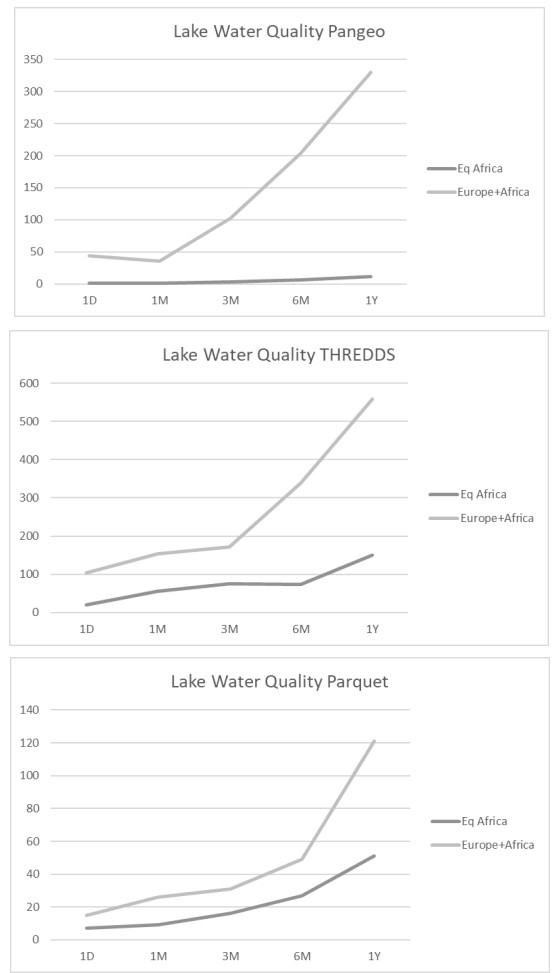

**Figure 14: Execution times in s for Lake Water Quality over periods of time**


The extraction for the Africa region is performed with 10 cores and with 50 cores for both Africa and Europe. The execution time of the extraction for the non-continuous data is greater than the global extraction for the oceanographic data with 50 cores (See Figure 12). In the Parquet case, with 100 cores, the result of the European and African coverage extraction is 5 times greater than the oceanographic data worldwide extraction performed with 50 cores. Even if the tested data is not a big amount

of volume, it is necessary to read a lot of chunks to extract it. These formats are not optimized for gridded data with a poor geographical coverage distribution, or the chunk size should be optimized for this kind of dataset.

### 4.3.3 Enrichment along track results

Note: The TDS services are not designed for such a use case since a call to the THREDDS service should be done for each location. Therefore the enrichment along track use case was not performed for these solutions.





For Pangeo-Zarr, the results are presented as the mean execution time of 3 executions of each request.

The dataset used for the enrichment is global-analysis-forecast-phy-001-024 daily.

Tests are run with 10 cores of 4Gb, except for the incidence of the number of cores.

**Incidence of the number of positions in an area**

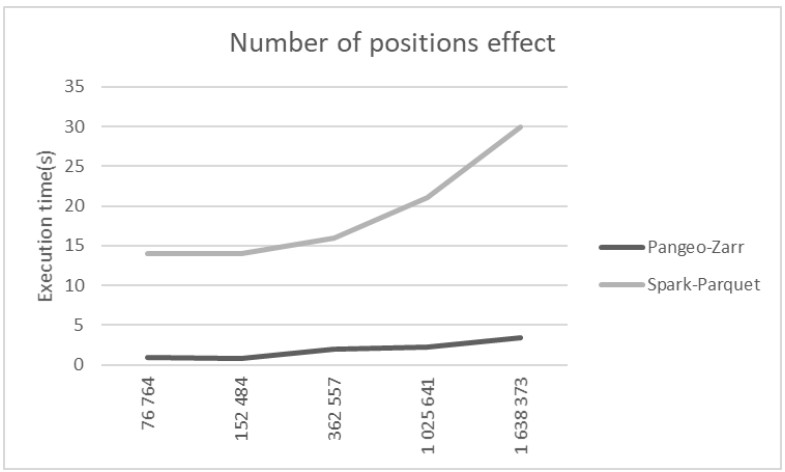


**Figure 15: incidence of number of positions for enrichment along track**

The Pangeo-Zarr implementation is more performant and more linear than the Spark-Parquet one when the number of locations is increasing. Nevertheless the extraction of 1,6 million positions in 30 seconds for the Spark-Parquet implementation is still

a good result. The enrichment with THREDDS would require a prohibitive execution time since it would imply a request per location.

**Incidence of the geographical coverage**

Comparing enrichments of positions in a local area or worldwide is another topic to consider: for the enrichment of 152484

position in Rotterdam and of 176735 positions in the world, the execution time is quite constant for the Pangeo-Zarr implementation (respectively 0.80s and 0.82s) but is 4 times more for the Spark-Parquet implementation (57s instead of 14s).

**Incidence of the temporal coverage**

The temporal coverage from 1 to 50 days includes a significant increase in the number of positions:




| time | nb points |
|------|-----------|
| 1D | 176 735 |
| 5D | 876 941 |
| 10D | 1 736 849 |
| 20D | 3 588 894 |
| 50D | 8 897 349 |

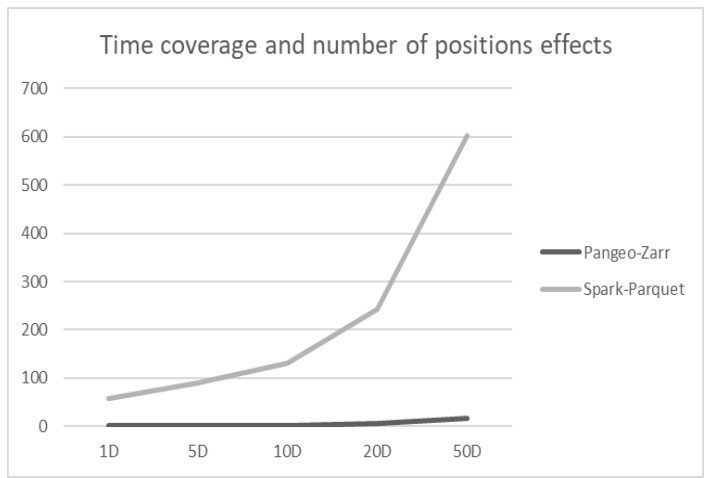


**Figure 16: incidence of the temporal coverage for enrichment along track**

The time coverage and number of positions increase is well supported by the Pangeo-Zarr implementation, less for the Spark-Parquet one which seems to be parabolic.

**Incidence of the number of variables**

Runs were executed with the following requests for positions around the world.

- Number of variables from :
- o 1 3D variable : sea_surface_height_above_geoid
- o 7 variables :
  - ocean_mixed_layer_thickness_defined_by_sigma_theta,
- sea_water_potential_temperature,
  - eastward_sea_water_velocity,
  - northward_sea_water_velocity,
  - sea_water_salinity,
  - sea_water_potential_temperature_at_sea_floor,
- sea_surface_height_above_geoid

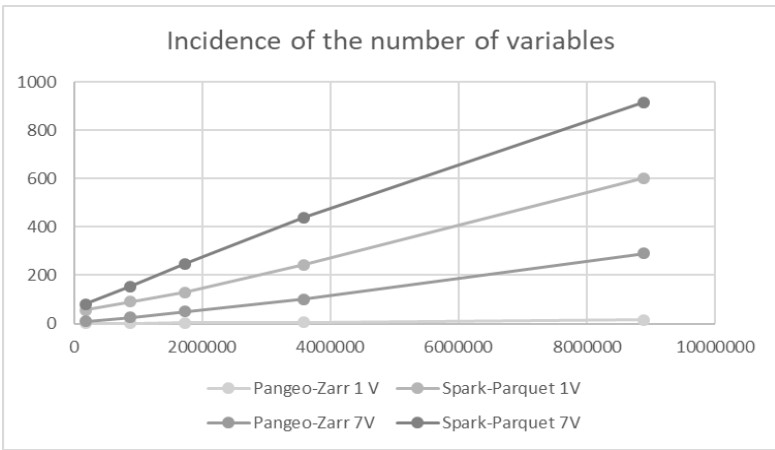

**Figure 17: Number of variables incidence for enrichment**





For the Pangeo-Zarr implementation, the mean ration between the execution times from 1 to 7 variables is 16 times slower only 1.6 for the Spark-Parquet implementation which is more time consuming.


**Incidence of the number of cores**

The increase of cores used to perform a worldwide extraction over 50 days for 8 897 349 positions provides the following result:

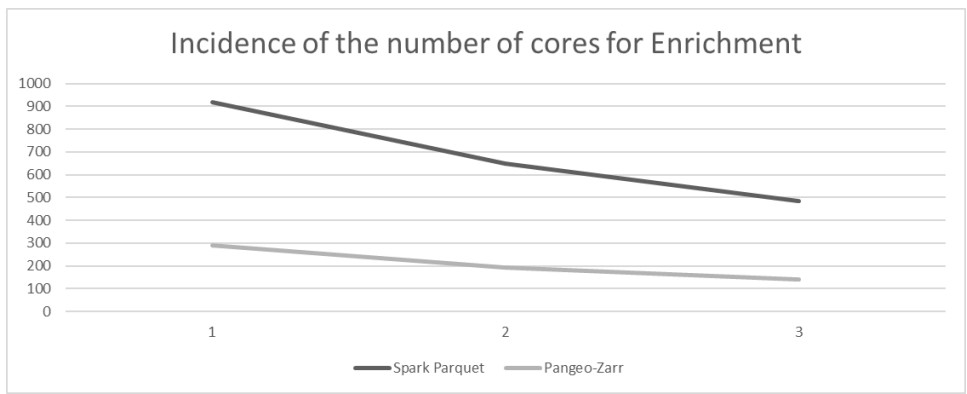

**Figure 18: Number of cores incidence for enrichment**

The scalability of the big data architecture enables a significant improvement for both implementations, but the limits were not found and a test with 50 or 60 cores could be set to try to find it.

**4.3.4 Parallel processing results**

The Jupyter Notebook in the CNES HPC is used to run the process of the sea level yearly mean, and its conversion in a 530 Spark/Scala job is run in Zeppelin notebook on the CLS big data architecture.

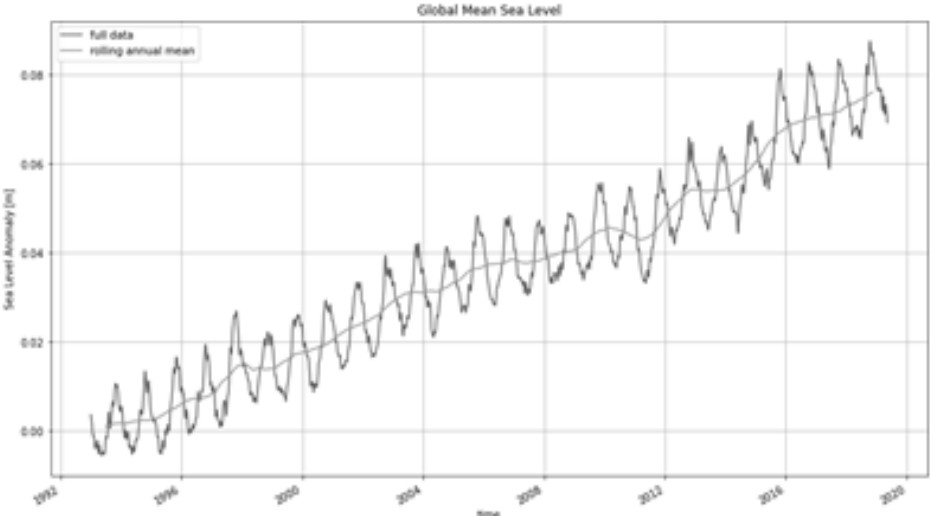

**Figure 19: mean seal level**

The mean execution time over 5 runs is 38 s for Spark/Scala with 50 cores and 11 s for Dask with 40 cores. The overhead 535 between Spark/Scala and Dask is due mainly to the initialization of the Spark context which is around 30s within the CLS architecture.





## 6 Discussion

### 6.1 THREDDS

In future versions, UNIDATA intends to make the partitioning strategy customizable by dataset (in order to avoid having too many small files when we can group data by month for example). While the initial S3 access supports a number of data access patterns, the work of this study has shown that there are issues with handling very large datasets and requests that result in very large response sizes. To better support large collections of data and reduce the need for rewriting existing datasets, the THREDDS team is working to implement support for virtual aggregation of datasets stored in S3 object stores.

The current study has suggested a number of possible items for future work:
- Requesting NetCDF-4 files with NCSS is RAM intensive and requires the resulting NetCDF-4 file to be written on the server before the server can start sending the HTTP response. The TDS CdmRemote Service can return a ncstream response which can start as the data is being retrieved. This would, to some extent, give a better indication of data access times by separating it from the time required to write the NetCDF-4 file.
- Further investigation is needed on caching of S3 reads in the TDS to see if changes might improve performance and memory usage.
- Once virtual aggregations of S3 datasets are supported, look for possible performance improvements.
- Longer term, the THREDDS team will look at how to parallelize reads from S3 object stores.

### 6.2 Pangeo-Zarr

The performances of the Pangeo solution are the best of the 3 compared solutions. The main limitation is that this solution is only accessible to the Python users.

### 6.3 Spark-Parquet

Several opportunities to optimize the performances are studied:
- Geoindexation
- Variables correlation
- Chunk size
- Scalable ingestion

The use of the quadtrees, geowave Hilbert curves or Google-S2 indexes can optimize geo extractions by zone, storing the chunks of partitions according to their geographical vicinity. Using a quadtree data structure could allow to build an index that associates each cell id to the location of the data chunk that contains its data tile.

Some scientific data have the particularity of being tightly linked, meaning that for most scientific use cases, these data are always accessed together. A notable example for climate data is wind or current data. In NetCDF files, they are encoded using 2 variables embedded in the same file: u, v. When processing or visualizing them, the user or the process accesses to both variables on the area and/or the period of interest. Their storage should be in the same chunk or closed chunks to avoid additional read access.




Then, a tuning could be performed to adjust the chunk size to the dataset hierarchy and size. in this study, the daily partitioning was suitable for the tested data but a specific test could have been performed to estimate the impact of the partition size. A parameter could be customized to adjust the dataset structure with the storage partition size. Additional tests will lead to recommendations according to datasets characteristics, but a classification of the dataset types should be studied first to help

users of the Parquet Cube solution to tune it.

Reducing the ingestion time by scaling the ingestion process is an easy step to be performed. This scalability is implemented in the operational environment at CLS premises to ingest files in parallel and speed up the ingestion of datasets with a long time coverage.

**7 Conclusion**

The Parquet format is a convenient implementation to open the data to a large community of users. Through a scalable process in a big data infrastructure, the ingestion of NetCDF data facilitates the processing of gridded data to extract easily the expected piece of information and make high added value processing chains or artificial intelligence modelling.

The presented results demonstrate that the Parquet format is compliant with the typical users'use cases in processing big

volume of gridded data: extraction of a subset of data to study or modelize, enrichment of positions with environmental variables, parallel processing to tune models is becoming accessible to scientists and compliant with operational processing requirements.

Alternatives could be runs of the benchmarks using Parquet/Dask/Python, TileDB or Cloud Optimize Geotiff formats for instance. The inputs and scenario provided here are detailed and free for anyone willing to compare such new implementations

of gridded data storage and access services in bigdata architecture.

**8 Data availability**

Gridded Data is around 2 Tb and is available on a NAS storage at CLS premises. The access can be provided on request to download data through HTTP.

| | | |
|---|---|---|
| 📁 dataset-duacs-rep-global-merged-allsat-phy-l4 | Dossier | 2020-04-24 09:26:56 |
| 📁 global-analysis-forecast-phy-001-024 | Dossier | 2020-04-21 18:33:04 |
| 📁 global-analysis-forecast-phy-001-024-hourly-t-u-v-ssh | Dossier | 2020-07-10 14:07:47 |
| 📁 LWQ100 | Dossier | 2020-04-24 09:53:41 |
| 📁 paper | Dossier | 2020-11-16 17:43:52 |
| 📁 tracks | Dossier | 2020-05-05 18:59:06 |

**Figure 20: NAS-EXT tree**

Please send a mail to bigdata4science@groupcls.com that will provide a slot time, an URL and password for downloads.

You can also perform the tests with the rolling year on line on the Copernicus services, but the availability of these datasets is not guarantee in the long term.

The along track dataset is less resource demanding and is available on Zenodo with the Digital Object Identifier

10.5281/zenodo.4894095, https://zenodo.org/deposit/4894095

The requests with all the parameters for subsettingand along track enrichment are available on Zenodo with the Digital Object identifier 10.5281/zenodo.4905665, https://zenodo.org/deposit/4905665

The source code ingestion is under L-GPL license and identified with the 10.5281/zenodo.4905679 DOI. It is available as an archive in Zenodo, https://zenodo.org/deposit/4905679 or in the Github repository

https://github.com/clstoulouse/parquetCubeIngestion



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
