# Peer review of "A Parquet Cube alternative to store gridded data for data analytics and modeling"

_Geoscientific Model Development, 2021_

## Author Comment (AC1)

The authors propose to use Parquet for storing gridded data efficiently. They compare NetCDF (using the THREDDS Data Server), NetCDF on S3, Pangeo-Zarr and Spark-Parquet using a number of different scenarios.

The comparison shown in the paper is interesting, but suffers from a few flaws: It is hard to compare the different results since the experiments use different hardware configurations and different compression algorithms. However, most notably, Pangeo-Zarr seems to perform best in majority of experiments (sometimes being orders of magnitude faster than Spark-Parquet) but the authors still propose using Spark-Parquet. Overall, the benefits of Spark-Parquet do not become clear. The paper would benefit significantly from more thorough explanations and uniform experiment configurations.

We first thank the reviewer for its thorough analysis of our paper and his comments.

The objective of the paper is to demonstrate that the Parquet Cube implementation is a good opportunity to process data on large volumes without being dependant on a language or technology, like it is the case for ZARR/Python, for some basic end users scenarios.

Several combinations could have been put in place, mixing Parquet with Python for instance. This paper is dedicated to people that are not in the Pangeo community, to tell them that the Parquet format is fine to develop performant services with Spark jobs (since it is also widely spread) or other means, even if the Scala/Spark implementation is not the best. This study is just giving a snapshot of what we put in place and displaying transparent results.

It was not possible to deploy the different implementations in the same infrastructure since the UNIDATA team was still developing the THREDDS service for the storage on S3 during the study. In fact, to determine what implementation is the best in term of performances, the same environment should have been used and reserved because even on a same cloud infrastructure, we cannot guarantee that resources in place are strictly the same due to the concurrent access to the storage by external users for instance, I/O can be the bottle neck. The effort to run several times all the requests proposed in the study was too high to be able to measure a precise execution time (running ten times, removing lowest and highest values to compute the mean) That's why this paper is focusing on describing the feasibility and quantitative results rather than providing a scientific comparison of performances between the different solutions.

However, we modified the manuscript to better stressed the main objective of the paper and addressed your comments

41: The citation format doesn't seem to adhere to the journal's guidelines (https://www.geoscientific-model-development.net/submission.html#references). This also applies to the following citations.

As it is mentioned, in the reference you provide, in terms of in-text citations, the order can be based on relevance, as well as chronological or alphabetical listing, depending on the author's preference.

We are using the alphabetical listing of references. We reviewed all citations to make sure we meet the guidelines now

Figure 1: All figures should be referenced explicitly in the text. This also applies to all other figures.

Done.

146: Why do you describe the system configuration here? It's also mentioned later.

**The system configuration for the first scenario is limited and does not use the full capacity of the configuration.**

182: The parallel processing scenario is not very detailed. How does it work? What happens in parallel? In general, all three scenarios could use some more explanations, for instance, showing their (pseudo) code.

The source code is given in a Pangeo notebook written by Abernathey https://github.com/pangeo-gallery/physical-oceanography/blob/master/01_sea-surface-height.ipynb. We added the reference here even if it was also mentioned in 91

192: The code for the different benchmarks should be provided to allow readers to compare/reproduce them. The Git repository only seems to contain code for converting NetCDF to Parquet.

Yes the paper promotes the Parquet Cube as a solution opened to different technologies and let users take advantage of this opportunity in their environment. Nevertheless, CLS provided the Python source code for the Pangeo implementation as notebooks in https://github.com/clstoulouse/ParquetCubeNotebooks .

CLS has IPR for the Spark services. Its deployment is quite complex and will need support. This support should be paid to perform the tests in another environment.

The TDS S3 capabilities were being actively developed. However, the source code was being pushed to the TDS GitHub repository (https://github.com/Unidata/tds) on the main development branch before it was used for testing.

193: How is the THREDDS implementation used, a Java application or something else? It does not become clear from the description.

Yes THREDDS tests are performed using a JAVA test application. We have enriched the description.

229: Pangeo-Zarr uses LZ4 compression. Does NetCDF also use compression in your experiments? If not, Compression can add significant overhead, how are you able to compare them fairly?

The NetCDF4 used during the bench is also implementing a compression. For the object store, no compression was used. Chuncks are in place without compression.

250: How does this scalable process work?

A docker is dedicated to the ingestion of a single file. Scaling horizontally the ingestion allows the parallelization of the ingestion. We added the following explanation in the manuscript: "multiple ingestion docker instances can process files in parallel, each of them ingesting a one of the input files."

Figure 3: What does this figure show? It does not become clear from the text. Moreover, the text in the figure is quite blurry and its resolution should be increased.

The storage tree displays for a dataset the content of the data folder including a parquet file per day and an index folder including the index files and versioned metadata. We made the figure more readable and added some comments.

301: Spark-Parquet seems to use Snappy, which has different performance characteristics than LZ4 used previously. For a fair comparison, all approaches should use the same compression algorithm (or none).

You are right but the difference between the default compressions of each implementation can be negligeable for the objectives of the paper. As stated in the paper, we made some compressions benches:

*Figure 1: compression performances in read access (in s)*

And as mentioned in the paper and described in the above figure, the difference between LZ4 and Snappy is not significant enough to change significantly the results. Keep in mind that our objective was not the optimization of performances but the possibility to use Parquet easily with a default configuration.

Figure 5: This figure shows that Snappy and LZ4 result in different file sizes, which in turn means that the benchmarks will have to read/write different amounts of data, possibly skewing the comparison.

As explained just above, the difference between LZ4 and Snappy is in fact introducing a bias but the objective of the paper was to promote the Parquet format and not to get a detailed tuning and optimized implementation. The default configuration is already compliant with the studied use cases.

330: Every scenario seems to run on different hardware, making it hard to compare the results. 512 GB vs. 12 GB RAM will have significant impact on caching. Pangeo-Zarr uses GPFS, while NetCDF is stored on local disks. GPFS is expected to be much faster than a local disk.

Absolutely right. We are aware of this. As mentioned, the objective is to provide a rough idea of performances with a default configuration of the solution, not to push it to limits. We have added clarity to the manuscript to better state the objective of the paper

350: How did you make sure they were using the same amount of memory? Did you also account for differences in CPU performance (apart from using the same amount of cores)?

We did not monitor the load average and memory usage during the bench. We considered that the operating system is providing the best performance with the available resources. We just checked the memory definition at the creation of the VM or the memory allocated to the process in the CNES HPC context.

 Table 2: Why is THREDDS-S3 larger than NetCDF? Aren't they both using NetCDF?

The size of the chuncks were not optimized in this purpose. An additional work could be done to fine tune it, but as the purpose of the study was not to get the ultimate performance, we didn't work on it. The size is fixed and closer now in the paper since the conversion from bytes to Tb introduced an error.

373: What are these numbers supposed to tell the reader?

It provides an idea of the duration to ingest the data if the reader wants to replay the bench. The parquet ingestion can be long.

387: Pangeo-Zarr and Spark-Parquet were also using NetCDF then? According to Table 2, all of their datasets were below 1 TB.

The preliminary idea was to provide to users the possibility to extract data in NetCDF whatever the implementation is, to give the user the output he is friendly with and compare the same outputs. It was not a good idea for two reasons:

- the generations of NetCDF outputs were too slow as the output size was growing
- It was preventing the user to access to a high volume of data to process and squeeze the advantages of the solution.

We removed this additional NetCDF conversion step which is was not meaningful for the study in the end and not the way to go to use big data volumes in the cloud.

395: What does "10 cores of 4 Gb" mean? 4 GB in total or 4 GB per core?

4Gb per core

Figure 9: The different scaling on these figures implies that both Pangeo-Zarr and Spark-Parquet had similar performances, but Pangeo-Zarr was significantly faster.

Yes, you are right, we adjusted the scale to distinct the lines on the graph. Otherwise, it is not possible to see the different lines.

- Figure 11: The different scaling makes it hard to compare results again.

Yes, you are right, we adjusted the scale to distinct the lines on the graph. Otherwise, it is not possible to see the different lines.

- Figures 15 and 16: These figures seem to imply that Pangeo-Zarr has much better scaling behavior than Spark-Parquet. Can this be explained somehow?

Unfortunately, we did not investigate further to check every configuration or parameter to explain the results in detail. The main objective of this paper was to demonstrate the possibility to use the Parquet format to store data and make it available and open to many technologies and tools.

- Figure 18: Why are you only using 1 to 3 cores here?

It was a axis legend error. It is corrected.

- 534: Why do you compare 50 to 40 cores? That makes it hard to judge the differences.

Yes, unfortunately the tests were done by different teams and we did not ask to replay them. The main idea is to see that the processing time for Parquet/Spark is not so long, even if it is higher than the Pangeo/Dask solution.

- 539: It seems NetCDF-S3 has several limitations. Have you also considered HDF5 (https://www.hdfgroup.org/solutions/enterprise-support/cloud-amazon-s3-storage-hdf5-connector/)?

No, we didn't switch to HDF5 storage. This paper was the opportunity for UNIDATA to stress their new THREDDS-S3 implementation still under development.

- 555: You mention that Pangeo-Zarr is limited to Python. Isn't THREDDS also limited to Java? Which languages are supported by Spark-Parquet?

THREDDS is in fact a JAVA library but the client to access the data using the opendap protocol can be relying on different technologies or tools (see 6.1 https://opendap.github.io/documentation/UserGuideComprehensive.html) . NETCDF libraries are also available in Python, https://unidata.github.io/netcdf4-python/. Apache Spark supports the following four languages: **Scala, Java, Python and R**

- 586: How do you support this conclusion? Pangeo-Zarr seems to be the fastest format and Python most likely has the largest community. Moreover, Pangeo-Zarr shows much better scaling than Spark-Parquet in a few of your experiments.

The objective of the manuscript is to explore and demonstrate that Parquet is an alternative format, compliant with typical and widespread scientists 'use cases and technologies. Sharing this information can help people who are dealing with different technologies by legacy or infrastructure constraints.

- 615: The last access dates for all references seem to be in French.

OK Done

Layout problems, typos etc.:

- 186: "4 go" - Is this supposed to be "4 GB"?

OK Done

- 209: "manor" - Should be "manner".

OK Done

- Figure 4: This figure is also blurry.

OK Done

- Figures 5 and 6: The axis descriptions are very hard to read due to the low resolution.
We resized the figures to make the X axis more readable

- 339: "Go" - Should be "GB". Also applies to the following lines.

OK Done

- Figure 11: The second row seems to be titled incorrectly, THREDDS-NC is missing and instead Spark-Parquet is shown twice.

Fixed

- Figure 19: This figure is so blurry that it's impossible to read.

Fixed

---

## Author Comment (AC2)

The authors perform a comparison of several systems on 3D and 4D gridded data, with a particular view on parallelization. Tools addressed are THREDDS/NetCDF on standard file system and an object store, pangeo/Dask, and Hadoop/Spark/Parquet. The conceptual model is a 4D cube where the vertical dimension is "nullable" in case of 3D data. This resembles the model of the Barrodale engine on top of Informix, for example.

In Section 2, I get puzzled about the experimental setup description:

 - the "criteria identified to stress" are not justified and sometimes surprising: why is it relevant for storage (!) whether a pixel is ocean or land? why is complex processing, heterogeneous data fusion, etc. not considered?

We first thank the reviewer for its thorough review of the paper and the detailed analysis and comments.

We are now clarifying in the manuscript the criteria and the choice of datasets.

 - "enrichment of CSV locations": CSV = comma-separated values? if so, why is that format choice important over, e.g., JSON? what does enrichment mean, and what is the scenario? why is it relevant?

Explanations of the enrichment and references are added to the manuscript

 - "parallelized processing" (check language use!) is not a user scenario, but an implementation detail. What would be interesting is what operations exactly to parallelize - for example, an edge filter or matrix operations are harder to parallelize than the trivial pixel operations such as NDVI or subsetting.

Statistics and trends detections are performed with parallelization. The idea is to prove the efficiency of the proposed storages, compared to traditional THREDDS solution.

Generally there seems to be a lack in concise description of the relevant choices, impacts, and measures. Just one example (p 11): "We repeated the tests several times to obtain the most reliable metrics." A rigorous approach might "run all tests 5 times on a [hot|cold] setup, discarding the maximum and minimum value and averaging the 3 remaining measurements".

The following precision was added: We repeated the tests three times to get a more reliable measurement of performances

The algorithms used for the scenarios, such as extracting significant continuous areas and enrichment, are only given cursorily, without a rigorous pseudo code or mathematical description.

Notebooks used to test the Pangeo implementation were added in github, https://github.com/clstoulouse/ParquetCubeNotebooks and referenced in the paper.

In 3.3, it is claimed that datacubes need reprojection which should be avoided. However, for join/fusion of datasets in different projections there needs to be a reprojection. And rescaling (which likewise involves resampling) is performed routinely by the approach. So I do not see the substantial difference. In fact, substantial preprocessing takes place at datacube creation time, massaging data to make them suitable for the processing later on. For example, all cubes are forced into the same space/time resolution - a brute-force method, and certainly more dangerous from a scientist perspective than reprojection. Other datacube approaches work on the original data and perform dynamic recombination.

Could you please precise why it is dangerous for a scientist to use a library to process the data the way he expects? It is true that the Rasdaman implementation makes it easier to access the data and that such ingestion mechanism facilitates the use of a language to manipulate the data. With this new Parquet datacube, we describe an alternative based on another paradigm.

Reading the datacube manifesto, https://earthserver.eu/tech/datacube-manifesto/The-Datacube-Manifesto.pdf, we can see that the Datacube terminology used for the Parquet datacube described here could not be compliant with the last requirement: "Datacubes shall support a language allowing clients to submit simple as well as composite extraction, processing, filtering, and fusion tasks in an ad-hoc fashion".

It is true that the Spark access to the Parquet Cube developed by CLS is not compliant with this 6th requirement of the datacube manifesto, but the Pangeo implementation which allows the SQL like selection in a dataframe initialized from the parquet files makes the Parquet Cube complaint with it.

The Parquet format does not seem efficient for cubes. By materializing the coordinates for each point the data volume is blown up immensely, and processing (including data bus transport between RAM and CPU) get slowed down. Combine this with the 3x explosion contributed by HDFS (not to speak about its inflexible page size), it is not clear how this approach can be efficient in comparison to others published. Certainly not a "green computing"!

The results of this paper are comparing the native NetCDF files size with the Parquet Cube implementation. We did not see the volume blowing up immensely doing our research.

Partitioning, a well-known technique for gridded data, is applied here as well. The parameters chosen are not justified, though: why regular partitioning? why partition length 1 day along time? We just learn "the partitioning-per-day makes sense",probably because the demonstration scenarios have been pre-trimmed to that. This will be problematic in a general-purpose operational deployment.

You are right, we did not stress the Parquet Cube implementation with the partition size and this limitation is mentioned in 595

Sadly, the performance comparison of the different implementations was done on rather different infrastructure, so comparison of the results is problematic.

This limitation is now clarified and commented in the manuscript. The objective of this paper is to explore and demonstrate that the alternative using the Parquet format has some strengths and weakness compared to others but nonetheless interesting for specific users and use cases.

In Table 3, performance results for single-file conversion are reported. There is a breathtaking span from 1 to 12816 seconds per file. Unfortunately, it is not explained sufficiently.

Indeed some measurements were not performed for all datasets and implementations that is why there are not applicable (N/A) values present in table3..The input files were too big for the THTREDDS implementation and unfortunately, the Pangeo implementation did not log the ingestion time. We do not think it was relevant to detail in the paper as it does not change any of our conclusion or hypotheses.

Performance results (Figure 8) are a little hard to follow due to nonlinearity - eg, time axis extraction starts with 1D (incidentally the stored grid resolution) and then scales with a factor 30 (?), 3, 2, 2. Equidistant spacing would help understanding the results.

Comment are added:

    In figure 8 below, time units are D for days, M for months, Y for year.

    The linearity in time for a given geographical region can be observed for the Pangeo/Zarr or Spark parquet implementations, except when the initialization of the context request of the subsetting is dimensioning compared to the subsetting process itself.

Surprisingly, performance degrades quadratic with increasing data volume returned, whereas other tools in the field commonly show a linear behavior. Parallelization does not seem to help much.

Unfortunately, we did not find yet the explanation for this quadratic effect.

Also surprising (and unexplained) is the non-monotonicity in Fig 11 for THREDDS / North Sea while THREDDS / global shows reasonable monotonicity. Further, in Fig 13 there is a huge outlier for THREDDS / hourly / 1M - is this a measurement issue, or a real result? Unfortunately, no explanation is attempted.

The stride bias mentioned in 454 is described in 255. We did not repeat it in the aforementioned paragraph.

On p 18 I would like to understand how "significant" data are determined algorithmically - as it stands, the system load generated cannot be estimated. Further, as "continuous" areas are retrieved there is likely some kernel operation involved. How does kernel computation work at partition boundaries, is it still correct (ie, fetches values from neighboring partitions where necessary)?

Significant means here with a physical value not a default value. We made the following clarification: The idea was to compare the incidence of the data type: significant continuous

physical values over wide areas (seas) versus lot of small data spots and long series of default values (lakes).

Looking at the results on p 19: Performing a simple subsetting returning an estimated 5 MB of data using 10 cores in 30 seconds is breathtakingly slow - other systems can do that in less than 1 second.

The 30 second measurement is mainly due to the initialization of the Spark Context. It was mentioned in 535 but you are right, we have also clarified this in this paragraph of the manuscript.

Overall, seeing the data set in the conclusion is described as having 2 TB and Spark/Dask were used on 50/40 cores: that means about 40 GB per node - loading that into RAM of each node and using just numpy etc. should be faster by orders of magnitude. Shouldn't the test go well beyond the cumulated RAM?

Sorry, I think there is a confusion here. The typical configuration used for the scenario 1 in the CNES HPC for Pangeo is 10 cores and 4 Gb of memory as mentioned. The tests were run in an environment quite easy to reproduce for readers and not expensive.

Bottom line, what I take home is

- THREDDS is not really scalable (which is confirmed by other studies)

- Parquet works well in stuations where data and scenarios are carefully aligned

- benchmark deployments have so many special tweaks and differences that a comparison is difficult

- both Spark-Parquet and Pangeo-Zarr fall significantly behind the performance of other tools around

- dynamic partitioning (as studied by Paula Furtado, for example) has not yet found wide recognition

OK. We updated the paper and the conclusion to focus on the feasibility aspect more than on performances.

editorial comments:

- p 3: URLs in the make the text ugly to read, better make it a reference.

OK, done

- p 3: "N variables" - what does that want to tell us? Unknown, variable, or...?

Ok, the numbers of variables are now mentioned in the paper.

- maybe recheck for typos, such as p 4 "librairies"
Fixed

- best use uniform nomenclature, not "4G" and "4 go" for 4 GB

OK done

- p 5: "The number of cores is increased for the Pangeo-Zarr and the Spark-Parquet environments." ...why? what is the number of cores there? Best motivate such decisions.

This motivation have been added to the manuscript : "to observe their horizontal scalability"

- Figure 9: as the diagram lines are greyscale they are not easy to distinguish. If color is not possible consider dashed lines etc.

Bold police and number format were adjusted to make the figures more readable.

- check for French words occurring, such as "Novembre"

Done

**Citation**: https://doi.org/10.5194/gmd-2021-138-RC2